# Insights into the Structure–Property–Activity Relationship of Zeolitic Imidazolate Frameworks for Acid–Base Catalysis

**DOI:** 10.3390/ijms24054370

**Published:** 2023-02-22

**Authors:** Maria N. Timofeeva, Valentina N. Panchenko, Sung Hwa Jhung

**Affiliations:** 1Boreskov Institute of Catalysis SB RAS, Prospekt Akad. Lavrentieva 5, 630090 Novosibirsk, Russia; 2Department of Chemistry and Green-Nano Materials Research Center, Kyungpook National University, Daegu 41566, Republic of Korea

**Keywords:** zeolitic imidazolate frameworks, acid–base properties, structural impact, chemical composition impact, particle size effect, catalytic properties

## Abstract

Zeolitic imidazolate frameworks (ZIFs) have been extensively examined for their potential in acid–base catalysis. Many studies have demonstrated that ZIFs possess unique structural and physicochemical properties that allow them to demonstrate high activity and yield products with high selectivity. Herein, we highlight the nature of ZIFs in terms of their chemical formulation and the textural, acid–base, and morphological properties that strongly affect their catalytic performance. Our primary focus is the application of spectroscopic methods as instruments for analyzing the nature of active sites because these methods can allow an understanding of unusual catalytic behavior from the perspective of the structure–property–activity relationship. We examine several reactions, such as condensation reactions (the Knoevenagel condensation and Friedländer reactions), the cycloaddition of CO_2_ to epoxides, the synthesis of propylene glycol methyl ether from propylene oxide and methanol, and the cascade redox condensation of 2-nitroanilines with benzylamines. These examples illustrate the broad range of potentially promising applications of Zn–ZIFs as heterogeneous catalysts.

## 1. Introduction

Over the last several decades, metal–organic framework chemistry has received considerable attention from researchers. Through active research in this field, more than 20,000 structures have been described to date; two-thirds of these structures are formed by carboxylate linkers. Moreover, a number of MOFs with zeolitic architectures were successfully synthesized as hybrid frameworks. Among them, zeolitic imidazolate frameworks (ZIFs) represent a new and special subclass of MOFs. ZIF structures comprise imidazolate linkers and metal ions (Co^2+^, Cu^2+^, and Zn^2+^) and are similar to zeolite framework topologies in which all tetrahedral atoms are transition metals, and all bridging atoms are imidazolate (IM) units. The first ZIF compound was [Co_5_(IM)_10_·2MB]_∞_, which is formed by imidazole and 3-methyl-1-butanol in the coordination sphere [1]. Currently, >200 three-dimensional ZIF structures have been prepared. ZIFs possess the characteristics of MOFs and zeolites. Thus, they are highly porous compounds characterized by extremely large specific surface areas (400–2000 m^2^ g^−1^), pores in the nanometer range (2–30 Å), high crystallinity, and exceptional thermal and chemical stabilities [2]. Many functionalities confer ZIFs with great promise in multiple application fields, including acid–base catalysis.

Due to the fact that the structure and physicochemical properties of ZIFs are similar to those of zeolites, one of the most important and commercially available catalytic materials, ZIFs and ZIF-based materials have been considered as promising materials for the catalysis. Multiple important points can be cited to support this assertion. Firstly, the combination of the corresponding beneficial characteristics of metal ions and IM linker catalytic properties of ZIFs can exceed the expectations for the simple mixture of their components.

Secondly, the high diversity of the building blocks of ZIFs enables their structural, textural, and physicochemical properties to change over a wide range. Thus, the structure of a ZIFs and the chemical composition of a linker can affect their affinity for the polar molecules (hydrophobic/hydrophilic properties). For example, ZIF-90 is a hydrophilic material because of the presence of the strongly hydrophilic group –CHO in its linker’s structure, whereas ZIF-8 and MAF-5/MAF-6 which have –CH_3_ and –C_2_H_5_ groups in their linkers, respectively, possess hydrophobic properties. This feature results in their superiority to zeolites in terms of catalytic properties in reactions involving polar reagents (water, alcohols, and aldehydes). Thus, in the condensation of benzaldehyde (BA) with ethyl cyanoacetate (EA), 76% BA conversion was achieved within 6 and 24 h in the presence of ZIF-8 and NH_2_-MCM-41, respectively [3]. A previous work [4] demonstrated that ZIF-8 was more active than the titanosilicate zeolite TS-1 in the condensation of formaldehyde with α-pinene. β-Zeolite was considerably inferior to ZIF-8 in the cycloaddition of CO_2_ to epichlorohydrin (EPH) at 343 K [5]. The zeolites TS-1, HY, and SBA-15 were even less active at 353 K. All the advantages of ZIFs are attributed to the blockage of active sites because of interactions via hydrogen bonds with the hydrophilic groups of zeolites. Thus, the reaction rate decreases in their presence.

Thirdly, the catalytic properties of ZIFs and those of zeolites [6,7,8,9] depend on particle size [10,11,12,13,14]. This phenomenon is unsurprising because an increase in particle size leads to a decrease in the external surface area of crystals and the appearance of diffusion hindrances for reagents. This phenomenon was demonstrated for most studied ZIFs such as ZIF-8 [11,12,13,14,15] and MAF-6 [12,13]. Despite their scarcity, these investigations indicated the possibility of tuning the rate and selectivity of certain processes by purposefully changing the external surface morphology.

Finally, ZIFs can be used in acid–base catalysis because of the presence of acidic sites and basic sites (BSs), simultaneously. ZIFs are active catalysts for transesterification [16,17]; the Knoevenagel reaction [2,18,19]; cascade processes such as the intermolecular variant of [3 + 3] cycloaddition [20,21]; the Friedländer reaction [19]; and organic carbonate synthesis [22,23,24,25,26]. Most of these processes proceed at the active site, which is the BS–Lewis acid site (LAS) pair [27].

Herein, we analyze the catalytic properties of ZIFs in terms of present insights into the structure–property–activity relationship. We primarily focus on the application of spectroscopic methods as instruments for the analysis of active sites. We provide a review summarizing the recent progress in the field of ZIF application in acid–base catalysis made by multiple research groups and ours in the recent 10–12 years

## 2. Tuning the Textural and Physicochemical Properties of ZIFs

First, a few words should be noted about the structure of the most studied ZIFs based on 2-methylimidazole (MIM) (ZIF-8, (Zn[MIM]_2_), 2-ethylimidazole (EIM) (MAF-5 (ZIF-14, (Zn[EIM]_2_); MAF-6 (ZIF-71, (Zn[EIM]_2_)); and imidazole-2-carboxyaldehyde (ICA) (ZIF-90).

The structures of ZIF-8, ZIF-90, and ZIF-91 are 3D frameworks with a zeolite sodalite (SOD) topology (Table 1) [28,29,30]. Their frameworks are formed by truncated octahedral (4^6^ 6^8^) cells connected by four- and six-membered rings of Zn_4_(2-RIM)_4_ and Zn_6_(2-RIM)_6_ (R: –CH_3_, –CHO) containing 24 Zn atoms per unit cell. This structure leads to large cavities (11.6 Å [ZIF-8] and 11.2 Å [ZIF-90]) interconnected by narrow holes (3.4 Å [ZIF-8] and 3.5 Å [ZIF-90]) [25,31].

A 3D framework of MAF-5 has analcime (ANA) zeolite topology with 4.0 × 5.8 Å apertures (Table 1) [29]. This framework is formed by cells with D3 (6^2^ 8^3^) symmetry that have elongated ellipsoidal cavities (7.0 × 10.0 Å) with windows of four-, six-, and eight-membered rings of Zn_4_(2-EIM)_4_, Zn_6_(2-EIM)_6_, and Zn_8_(2-EIM)_8_ [31].

The three-dimensional structures of MAF-6 and ZIF-11 have RHO topologies with large cavities (~18.7 Å) and apertures (~7.6 Å) (Table 1). The framework is formed by cells with windows of four-, six-, and eight-membered rings containing 48 Zn atoms per unit cell [32]. Each cell is connected to six adjacent cells by polyhedral units comprising double eight-membered rings.

### 2.1. Hydrophobic/Hydrophilic Properties

The topology and chemical composition of the linker can affect hydrophobic/hydrophilic properties. In fact, the hydrophobic and hydrophilic natures of several ZIF materials have been determined (Table 2).

ZIF-8 was reported to exhibit a strong hydrophobic character and remarkable hydrothermal stability. Furthermore, ZIF-90 was demonstrated to be a completely hydrophilic material because of the strong interaction of the aldehyde oxygen atom with water molecules via hydrogen bonds. We investigated the hydrophobic/hydrophilic properties of MAF-6 and ZIF-90 using IR spectroscopy. Samples were activated at 423 K under vacuum for 3 h. A broad band in the region of 2500–3550 cm^−1^ (dashed curves) and a band at 3646 cm^−1^ were observed in the spectra of pristine MAF-6 and ZIF-90 (Figure 1). These bands can be assigned to the ν(O–H) stretching vibration of adsorbed water molecules [37] and Zn–OH groups [16]. The addition of air to ZIF-90 and MAF-6 increased the intensity of the band at 3646 cm^−1^ (Figure 1). Note that the rate of change in the band intensity of ZIF-90 was higher than that in the band intensity of MAF-6. This phenomenon can be related to differences in hydrophobicity (Table 2). ZIF-90, which had a water contact angle of 71.1° [34], is a strongly hydrophilic material because it contains polar groups that are adsorption sites featuring hydrogen bonds. Furthermore, the surfaces of ZIF-8 and MAF-6 formed by 2-MIM and 2-EIM, respectively, were hydrophobic with large contact angles of 142° [35] and 143° [36], whereas MAF-5 formed by 2-EIM as MAF-6 was absolutely hydrophilic with a contact angle of 0° [34]. The effect of the functional groups in the linker framework on the affinity for polar molecules may be important for two reasons. Firstly, polar reagents (alcohols, aldehydes, and others) and water formed in the reaction can block active sites because of their interaction via hydrogen bonds with the hydrophilic groups of the catalysts and thus decrease the reaction rate. Secondly, polar groups can activate reagents because of this interaction.

### 2.2. Textural Properties

An analysis of the literature indicates that the textural properties of ZIFs depend on multiple factors such as the method and conditions of their synthesis, chemical composition, and particle size. Several examples that we obtained are provided below.

**Effect of chemical composition.** We recently demonstrated that the chemical composition, i.e., the atomic ratio of Co/Zn, affects the textural properties of mixed Zn/Co–ZIFs [38]. Replacing Zn^2+^ with Co^2+^ changes the specific surface area and microporosity. The increase in Co content magnified the specific surface area and contribution of the micropore volume to the total pore volume (V_μ_/V_Σ_) (Table 3). Thus, after the substitution of 25% of Co^2+^ ions with Zn^2+^ ions, the specific surface area and V_μ_/V_Σ_ decreased from 2058 to 2011 m^2^/g and from 1.00 to 0.96, respectively. The higher specific surface area (2058 m^2^/g) and (V_μ_/V_Σ_) (1.0) of ZIF-67(Co) than those of ZIF-8(Zn) can likely be explained by slight differences in frameworks, namely, cavities with a size of 11.6 Å accessible by small windows of 3.4 Å for ZIF-8(Zn) and cavities of 11.4 Å and small windows of 3.3 Å for ZIF-67(Co).

**Effect of particle size.** The particle size of ZIFs is one of the parameters that allow changing textural properties. The approaches to obtain ZIFs with different particle sizes deserve to be mentioned. As per many studies, the morphology and size of ZIF-8 crystals can be tuned by varying experimental conditions, e.g., the nature of the solvent or Zn source. Schejn et al. [19] demonstrated the synthesis of ZIF-8 with different crystallite sizes and morphologies in methanol using various salts, such as Zn(NO_3_)_2_, Zn(OAc)_2_, and ZnBr_2_ (Table 4), as the zinc source. The ZIF-8 crystals obtained from Zn(NO_3_)_2_ and Zn(OAc)_2_ were in the form of truncated rhombododecahedrons with an average size of 141 ± 48 nm and rhombododecahedrons with an average size of 500 nm, respectively. Furthermore, the application of ZnBr_2_ led to the formation of large nanocrystals (~1050 nm) in the form of cubes and rhombododecahedrons.

The variation in the solvent’s nature during the course of synthesis enables ZIFs with different particle sizes to be obtained. This phenomenon can be explained by the role of the solvent as a structure-directing agent via its inclusion in frameworks through noncovalent interactions. Thus, we reported that the particle size of MAF-6 was correlated with the variation in the hydrogen bond donation of solvents (α) [39] (Table 5) and decreased in the order of
MAF-6(L) (MeOH) > MAF-6(M) (EtOH) > MAF-6(S) (PrOH)

In all solvents, the crystal morphology was a rhombic dodecahedron. Bustamante et al. [40] demonstrated a similar solvent effect on the nanocrystal characteristics of ZIF-8. Indeed, the textural properties of ZIFs depend on particle size. As shown by the data in Table 5, in MAF-6 the specific surface area and microporosity (V_μ_/V_Σ_) increased with the enlargement of the crystal size. These changes can be attributed to the effect of alcohols on the rate of nucleation and crystal growth and therefore impact the formation of pore structure. Similar correlations were demonstrated by the characteristics of ZIF-8 (Table 5, Figure 2A): in ZIF-8, the external surface decreased, whereas microporosity increased with the increase in crystal size [14,41,42].

### 2.3. Nature of Active Sites

Chizallet et al. [16] were the first to examine the nature of ZIF-8 active centers. The combination of IR spectroscopy using CO as the probe molecule and density functional theory (DFT) to calculate model structures suggested that ZIF-8 can have high-coordinated Zn^2+^ ions (four linkers in the coordination sphere, Zn–Im_IV_) and low-coordinated Zn^2+^ ions (without one [Zn–Im_III_] or two [Zn–Im_II_] linkers) in the framework (Figure 1). LASs are formed by low-coordinated Zn^2+^ ions (without one [Zn–Im_III_] or two [Zn–Im_II_] linkers). Brønsted acid sites (BASs) are formed by the –NH groups of the linker, and the BSs are Zn–OH groups and free N fragments belonging to the linkers. Recently, Lee et al. [26] examined the nature of the active sites of ZIF-8 synthesized in water and methanol solutions using FTIR and XPS. They found that ZIF-8 possessed pyrrolic and pyridinic N fragments and hydroxyl groups (Figure 1). Below, we provide several examples that demonstrate the effect of chemical composition and particle size on their number.

Effect of chemical composition. The effect of chemical composition on the number of active sites was recently demonstrated for mixed Co/Zn–ZIFs [38]. It was investigated by using DR–UV–vis spectroscopy. A band at 225 nm assigned to π→π* transitions in imidazole rings and a weak broad band in the region of 240–350 nm assigned to the ligand-to-metal charge transfer (LMCT) band [43] were observed in the DR–UV–vis spectrum of ZIF-8. The insertion of Zn^2+^ ions by Co^2+^ ions led to the intensification of the LMCT band. The intensities of these bands rose with the increase in the Co content in the framework of mixed Zn/Co–ZIFs. On the basis of DFT calculations [16,44], the bands at 252 and 295 nm were attributed to the LMCT from low-coordinated M^2+^ (Co-Im_III_ and Co-Im_II_) to ligands and high-coordinated M^2+^ atoms (Co-Im_IV_) to ligands, respectively. Increasing the Co content in the framework of mixed Zn/Co–ZIFs elevated the I_252_/I_295_ ratio of band intensities (Figure 2B), i.e., the content of Co-formed active sites (Co-Im_III_ and Co-Im_II_) increased with the increment in the Co content in the structure of Zn/Co–ZIFs. This phenomenon can be explained by the slight changes in structure and the difference between the ionic radii of Zn^2+^ (0.076 ± 0.002 nm) and Co^2+^ (0.074 ± 0.002 nm).

**Effect of particle size.** Particle size is one of the important factors affecting the number of active sites. The effect of particle size on the nature and number of active centers was studied in detail for MAF-6 [12,13]. The analysis of MAF-6 basicity via IR spectroscopy using deuterated chloroform (CDCl_3_) as a C–H acid probe (DRIFT–CDCl_3_) demonstrated that the intensity of the 2242 cm^−1^ band (I_2242_), which characterizes the interaction of CDCl_3_ with the BSs of MAF-6, decreased with the increase in crystal size (Figure 2C–D). The DR–UV–vis data indicated a change in the nature of the active sites. Two absorption bands at 300 and 360 nm, assigned to the LMCT from low-coordinated Zn^2+^ cations (Zn–Im_III_ and Zn–Im_II_) to the ligand and from high-coordinated Zn^2+^ cations (Zn–Im_IV_) to the ligand, respectively, were observed in the spectra of all MAF-6 samples. The ratio of the intensities of these bands (I_300_/I_360_) can serve as a measure of the changes in the amounts of Zn–Im_III_ and Zn–Im_II_ in the sample, i.e., the changes in the amount of LAS. The decrement in the I_300_/I_360_ ratio of band intensities with the growth of MAF-6 crystal size may indicate a decrease in the number of LAS. The reduction in the number of BS and LAS with the increase in particle size was consistent with the change in D_SEM_/D_XRD_ ratio (Figure 2C), which can be considered as a parameter that reflects the relationship between regular and imperfect structural regions in the bulk of the MAF-6 particles. Figure 2D shows a good linear correlation between the change in the number of LAS (I_300_/I_360_) and BS (I_2242_) that is unsurprising because they are formed by bond breakage between the Zn^2+^ cation and the *N*-imidazolium ring atom.

Similar trends were demonstrated for ZIF-8. As per DRIFT–CDCl_3_, ZIF-8 possessed two types of BS that exhibit two bands at 2253 (strong BS) and 2245 cm^−1^ (weak BS) in the spectrum of adsorbed CDCl_3_ (Figure 2A). ^1^H NMR (MAS) data and DFT calculations indicated the presence of two types of main centers in the structure, which were in the micropores next to the –CH_3_ group and the C–H bond of the imidazolium ring [45,46]. DRIFT-CDCl_3_ data indicated that particle size affected the I_2245_/I_2253_ ratio of band intensities. The increase in particle size increased the I_2245_/I_2253_ ratio. Despite the increase in the number of strong BSs, their accessibility to the reagents decreased, as can be seen by the increase in the microporosity of the system (V**_μ_**/V_Σ_) (Figure 2A).

### 2.4. Basicity of ZIFs

Information on the basicity of ZIFs is limited and contradictory. Only two methods are currently used for examining the basic properties of ZIFs. Herein, we present the results obtained by these methods.

**Temperature-programmed desorption of CO_2_ (CO_2_–TPD).** ZIF-8 was studied in detail by using CO_2_–TPD [47,48,49,50,51]. According to Ahn et al. [48], ZIF-8 has two types of BSs, i.e., weak BSs (with desorption temperatures of 323–478 K) and strong BSs (with desorption temperatures of 450–673 K) (Table 6). The number of BSs depends on different parameters. Thus, according to Lai et al. [51], the increase in the molar ratio of Zn(NO_3_)_2_/2-MIM/MeOH from 1:7.9:86.7 to 1:7.9:1002 reduced the number of BSs from 0.70 mmol/g to 0.32 mmol/g because the particle size of ZIF-8 decreased from 250 to 40 nm.

The type of solvent affects the number of BS. Table 6 shows that the numbers of BSs of ZIF-8 samples synthesized in water with a small amount of triethylamine (TEA), MeOH, and dimethylformamide were 5.64–8.05, 0.7, and 0.1 mmol/g, respectively. We can suggest that these changes are related to interactions between the molecules of the solvent and BSs. Although the increase in the molar ratio of TEA/Zn(NO_3_ + 2-MIM) from 0.001 to 0.004 elevated the number of BSs from 5.64 to 8.085 mmol/g, excess TEA during synthesis reduced the number of BSs (7.17 mmol/g) [49].

Mousavi et al. [52] synthesized two ZIF-8 samples through solvothermal (SV) and spray-drying (SP) methods. CO_2_-TPD and NH_3_-TPD demonstrated that ZIF-8(SP) had more acidic sites and BSs than ZIF-8(SV) (Table 6).

**DRIFT–CDCl_3_**. This method was successfully used to analyze the basicity of metal–organic frameworks, such as MIL-100(Al); Cu(BTC)_2_; UiO-66 and NH_2_–UiO-66 [27,53]; and MIL-101(Al)-NH_2_ [54], as well as ZIFs [12,13,14]. The strength of BSs determined by this method is shown as proton affinity (PA) in Table 7. The data presented clearly demonstrated that the basicity of ZIFs was lower than that of linkers. Thus, the basicity of ZIF-8 (858 kJ/mol) was lower than that of 2-MIM (963.4 kJ/mol) [55]. A similar distinction was characteristic of MAF-6 (870 kJ/mol) and MAF-5 (884 kJ/mol) vs. 2-EIM (978 kJ/mol) [55]. This difference was attributed to the formation of a Zn–N bond via the interaction between Zn^2+^ and the nitrogen atom of the organic linker. The basicity of ZIF-8 was lesser than that of MAF-5, MAF-6, and ZIF-90. This difference can be explained by two reasons. The first is the lower basicity of 2-MIM than that of 2-EIM. The second is the difference in the pK_a_ of the conjugate acid for 2-MIM (7.85) [56,57], 2-EIM (8.0) [56,57], and ICA (pK_a_ 11.48 ± 0.10, Predicted) [58]. By comparing the basicity of the materials formed by 2-EIM, we reported that the basicity of MAF-5 was higher than that of MAF-6. The high basicity of MAF-5 can possibly be attributed to the high distortion of the framework of MAF-5 (Table 1), which can lead to changes in electron density at BSs.

## 3. Catalytic Properties of ZIFs from the Perspective of the Structure–Property–Activity Relationship

In this section, we discuss several reactions in which some understanding of the catalytic action of ZIFs in terms of the structure–property–activity relationship has already emerged. We primarily focus on condensation reactions (the Knoevenagel condensation and Friedländer reactions), the cycloaddition of CO_2_ to epoxides, the synthesis of propylene glycol methyl ether from propylene oxide (PO) and methanol, and the cascade redox condensation of 2-nitroanilines with benzylamines. For these reactions, efforts are already being made to formulate certain approaches and orientations for designing new catalysts based on ZIFs.

### 3.1. Condensation Reactions

In this section, we start with the Knoevenagel condensation reaction, which is one of the most studied reactions (Figure 2). From the perspective of an organic chemist, this reaction is of great importance because it enables a wide range of compounds with biological activity to be obtained. Based on the literature, various catalytic systems, for example, amine-functionalized mesoporous zirconia [59], basic MCM-41 silica [60], zeolite-exchanged alkylammonium cations [61], and MOFs [27,62], have been applied to this reaction. The use of ZIFs was reported [18,63,64]. Their application has several advantages. Firstly, the use of ZIFs allows the reaction to be performed under mild conditions at room temperature. ZIFs are superior to MOFs in terms of activity. Furthermore, in their presence, the reaction is heterogeneous, which facilitates separating the catalyst from the reaction mixture and reusing it.

The catalytic properties of ZIFs with different structures and chemical compositions in this reaction were examined. Nguyen et al. [18] examined the effect of the Zn–ZIF structure on the reaction rate of the condensation of benzaldehyde (BA) with ethyl cyanoacetate (ECA, R = CO_2_Et) in the presence of three samples, namely ZIF-8 (Zn[MIM]_2_, SOD), ZIF-9 (Co[IM]_2_, SOD), and ZIF-10 (Zn[bIM]_2_, MER). The reaction rate was correlated with the pore size of ZIFs, i.e., the largest sphere that will pass through the pore (d_p_) in the framework of ZIFs, and decreased in the following order [28]:ZIF-10 (8.2 Å) > ZIF-8 (3.4 Å) > > ZIF-9 (2.9 Å) 

Schejn et al. [19] demonstrated that particle size affected the activity of ZIF-8 in the condensation of 4-bromobenzaldehyde (Br-BA) with malononitrile (Table 4). The conversion of Br-BA within 30 min of the reaction decreased in the order corresponding to the increase in particle size:ZIF-8(141 nm) (96%) > ZIF-8(500 nm) (82%) > ZIF-8(1050 nm) (79%)

This order can be attributed to the decrease in the number of accessible active sites because of the decreasing external surface area of the ZIF-8 particles.

The size effect was shown in the condensation of 2-aminobenzophenone with acetylacetonate (the Friedländer reaction) (Figure 3) [19]. An increment in particle size increased the reaction rate and the yield of the 2-quinoline derivative (Table 4). The maximum product yield of 96% was observed in the presence of the ZIF-8 (141 nm) catalyst with the smallest particle size. Due to the fact that the kinetic diameter of reagents, for example, 4-methylquinoline (7.3 Å) [65] is larger than the pore diameter (d_p_) of ZIF-8 (3.4 Å), we can presume that the reaction primarily proceeded at the outer surface. Therefore, the decrement in particle size should lead to an increase in the outer surface area and in the number of available active sites for reagents.

### 3.2. Synthesis of Propylene Glycol Methyl Ether from Propylene Oxide and Methanol

ZIFs were also successfully applied as catalysts for the synthesis of methyl ether of propylene glycol (1-methoxy-2-propanol, 1-MP) from propylene oxide (PO) and methanol (Figure 4). This reaction is of practical interest because 1-MP is used as a solvent and as an intermediate for the synthesis of monomethyl propylene glycol acetate and pesticide (S)-metolachor. Notably, the reaction selectivity depends on the nature of the active sites. In the presence of catalysts with acidic sites (BF_3_, H_2_SO_4_), primary alcohol (2-methoxy-1-propanol, 2-MP) is the major product, whereas for catalysts with BS or active sites formed by the LAS-BS pair, the major product is 1-MP (secondary alcohol).

Previous works [12,14,66] demonstrated the possibility of using ZIFs as catalysts. First, these studies reported that the application of ZIF-8, MAF-5, and MAF-6 could reduce the reaction temperature to 413 K–423 K (Table 8). Moreover, in the presence of ZIFs, the major product with 81.9–92.6% selectivity was 1-MP. The high reaction rate and selectivity for 1-MP were likely caused by the specific affinity for polar molecules (hydrophobic/hydrophilic properties) (Table 2) and the nature of acid–base sites. The analysis of experimental data indicated that the conversion of PO decreased in the following order:ZIF-8 > MAF-6 > MAF-5 

This order agreed with the change in the strength of the BS (Table 8) and was unsurprising. As per an investigation of the reaction mechanism [14,66], an active site formed by the LAS–BS (Zn^2+^–N) pair is necessary for the activation and subsequent transformation of reagents. The physical adsorption of PO and the dissociative adsorption of MeOH to MeO^–^ and H^+^ occur at this site. Thus, the higher the strength of BS, the higher the reaction rate.

ZIF structure is another important parameter that must be considered. The data shown in Table 8 illustrate that the selectivity for 2-MP decreases in the following order:MAF-6 (9.2%) > MAF-5 (7.4%) > ZIF-8 (6.2%) 

This trend agrees with the decrease in pore size (d_p_) in the framework of ZIFs [28] (Table 1):MAF-6 (7.6 Å) > MAF-5 (5.8 Å) > ZIF-8 (3.4 Å) 

This correlation can indicate that decreasing channel size leads to an increase in steric hindrance that does not favor the formation of the bulky 1-MP molecule.

Isaeva et al. [14] reported that the catalytic properties of ZIF-8 depend on particle size. The increase in particle size from 125 nm to 610 nm led to the decrease in the conversion rate of PO from 54.4% to 18.1% over 5 h of reaction. Furthermore, the selectivity for 2-MP decreased from 93.8% to 83.0%. Undoubtedly, these results were attributed to changes in the textural properties of ZIF-8 (Section 2.2, Table 5) and the nature of the active sites (Section 2.2, Figure 2A). As a result of an increase in microporosity (V_M_/V_y_), the number and accessibility of active sites for the reagents decreased, which led to a reduction in the reaction rate. Moreover, increasing microporosity reduced the selectivity for the bulky 1-MP molecule.

### 3.3. Cascade Redox Condensation of 2-Nitroanilines with Benzylamines

The synthesis of various benzimidazole derivatives has piqued the interest of researchers because benzimidazole nuclei are part of the structures of many bioactive heterocyclic compounds. Two general methods for the synthesis of benzimidazole moieties are known, such as (a) the coupling of 1,2-phenylenediamines with aldehydes, acids, or esters and (b) the coupling of 1,2-phenylenediamines with amines. Nguyen et al. [67] provided the first report on synthesis via the combination of 1,2-phenylenediamine with amines (Figure 5) in the presence of cobalt or an iron halide under solvent-free conditions at 393–413 K.

An important role of Co is Co^2+^/Co^3+^ transformation in the course of the reaction. The reaction mechanism consists of several steps (Figure 6) [67]:

(A).Two of the initial steps are based on the redox transformations of the Co ions of the catalysts.
-Step of the oxidation of benzylamine. This stage comprises the oxidation of benzylamine (2) into benzaldimine, which is rapidly transformed into *N*-benzylbenzaldimine (4) via a transamination reaction.-Step of the reduction of an –NO_2_ group. In this stage, the –NO_2_ group of 2-nitroanil (1) is reduced into a product (3).(B).The following steps are based on the acid–base properties of the catalyst and include the formation of 2-phenylbenzimidazole (7) through a cascade condensation of (4) with (3) to form (5) and then (6).(C).The final step is the oxidation aromatization from (6) to (7)

Recently, cobalt-based ZIF materials, such as ZIF-4, ZIF-67, and ZIF-9, were used as catalysts for the synthesis of 2-phenylbenzimidazole via the redox condensation cascade of 2-nitroanilines with benzylamines (Figure 3) [68]. The yield of various benzimidazole derivatives was reported to be 67–99% in the presence of ZIF-67.

The evidence from this study suggested that various factors affected the catalytic behavior of ZIFs. Furthermore, the structure and chemical composition of ZIFs allowed the tuning of their catalytic properties. Thus, the reaction between the 2-nitroaniline and benzylamine activity of Co–ZIFs decreased in the following order (Table 9):ZIF-67 > ZIF-9 > ZIF-4 

This order was in line with the difference in the basicity of linkers (PA) [55] that decreased in the order of
MIM (963.4 kJ/mol) > bIM (953.8 kJ/mol) > IM (942.8 kJ/mol)

However, the textural properties of Co–ZIFs should also be considered. That is, the yield of 2-phenylbenzimidazole is adjusted by the pore size (d_p_) and the diameter of the largest sphere (d_L_) that will fit into the framework [28]. The yield of 2-phenylbenzimidazole decreased with reduction in d_p_ and d_L_:ZIF-67 (3.4 A, 12.3 Å) > ZIF-9 (2.9 A, 11.6 Å) > ZIF-4 (2.5 A, 4.7 Å)

### 3.4. Cycloaddition of CO_2_ to Epoxides

The use of carbon dioxide, whose reserves in nature are almost inexhaustible, as a feedstock for the production of cyclic carbonates is another example in which ZIFs have shown themselves to be promising catalytic materials [69,70,71,72]. The cycloaddition of CO_2_ to epoxides enables compounds (Figure 7) to be obtained that are widely employed as intermediates for the synthesis of polymers in various applications, including those in electronics, optical media, the automotive industry, and the medical industry.

First of all, we would like to focus on the situation in the field pertaining to the investigation of catalytic behavior in the cycloaddition of CO_2_ to PO and EPH. These reactions are the most studied and have revealed several approaches for the design of new ZIFs for the reaction of CO_2_ with other epoxides.

Toyao et al. [73] investigated the reaction of CO_2_ with epoxides in the presence of Zn- and Co-formed ZIFs with an SOD topology. They focused on the materials obtained after ZIF calcination at 873, 1073, and 1272 K. We were able to establish several trends. In the presence of Co–ZIFs and Zn–ZIFs, the yield of cyclic carbonate depended on pore size (d_p_) (Figure 4). Thus, ZIF-8 was more active than ZIF-7 because it had the largest d_p_ value (3.4 Å vs. 2.9 Å). A similar difference was observed between ZIF-67 (3.3–3.4 Å) and ZIF-9 (2.9 Å). The comparison of samples with equal d_p_ provided controversial results. The comparison of ZIFs with a d_p_ of 2.9 Å indicated that product yield was higher in the presence of the Zn–formed ZIFs than in the presence of Co-formed ZIFs, whereas the opposite trend was observed for ZIFs with a d_p_ of 3.4 Å.

Kuruppathparambil et al. [74] investigated the effect of Co on the activity of ZIFs in the cycloaddition of CO_2_ to EPH. The conversion of EPH was reported to depend on the Co content in the framework of mixed Zn/Co-ZIF and decreased in the following order:ZIF-8 (98.2%) > Zn/Co-ZIF (2:1) (96%) > Zn/Co-ZIF (1:1) (94.8%) 
> Zn/Co-ZIF (1:2) (80.2%) > ZIF-67 (66.5%) 

The selectivity for cyclic carbonate in the presence of Co/Zn–ZIF (90–98%) was higher than that in the presence of ZIF-8 (33.4%). Furthermore, other research groups reported that the conversion of EPH was slightly lower (65.4%) in the presence of ZIF-8 than in the presence of Co/Zn–ZIF (Co:Zn = 1:1) (69.4%) and that the selectivity for chloropropene carbonate (CPC) was 98–99% in the presence of the studied samples. Therefore, the effect of metals on the catalytic properties of ZIFs remains open-ended.

Recently, we investigated the catalytic properties of Zn- and Co-containing ZIFs in the cycloaddition of CO_2_ to PO in the presence of [n-Bu_4_N]Br as a co-catalyst at 353 K under 0.8 MPa of CO_2_ [38]. We found that the activities of Zn-containing binary systems were higher than those of Co–ZIFs (Figure 5A). Moreover, the activity of mixed Zn/Co–ZIFs decreased with the increase in Co content in the framework of ZIFs (Figure 5B) possibly because of several reasons:

Firstly, this reduction is attributed to the difference in the accessibility of active sites to reagents. This difference results from the slight differences in their frameworks. As per [29], ZIF-8 possesses a larger cavity (11.6 Å) and pore aperture (3.4 Å) than ZIF-67 (11.4 Å and 3.3 Å, respectively). Another reason may be the difference in microporosity (V_μ_/V_Σ_) (Table 3, Section 2.2). All of these differences can impact the accessibility of active sites to reagents and the adsorption and diffusion of reagents. The different numbers of active sites and strength of the Co-formed active sites in the interaction with reagents are other causes of the low activity of Co-containing ZIFs (Figure 2B, Section 2.2).

Nguyen et al. [75] reported that the activity of a Zn/Co-ZIF composite (1 mol Zn/1 mol Co) in the synthesis of cyclic carbonates from CO_2_ and epoxides can be improved through the partial substitution of 2-MIM by 3-amino-1,2,4-triazole (Atz) (Figure 8). The effect of the post synthetic modification of Zn/Co–ZIF with Atz for 0, 6, 12, and 24 h on catalytic activity in a reaction between CO_2_ and ECH was investigated. The Atz content in Zn/Co-ZIF was controlled by adjusting the ligand exchange reaction time of 2-mIM with Atz. The highest Atz content (32% of Atz) was present in the sample after 24 h of modification. The Atz content in Zn/Co-ZIF affected the conversion of ECH (Table 10). The increase in Atz content favored the increase in the conversion of ECH. Zn/Co-ZIF-Atz (24 h) possessed the highest activity, which resulted in 99.1% ECH conversion with 99% selectivity for CPC after 24 h without a cocatalyst and solvent under mild conditions (353 K, 0.1 MPa of CO_2_), in the presence of Zn/Co–ZIF-Atz (24 h). The change in catalytic properties was suggested to originate from the introduction of basic amine groups (–NH_2_) from Atz and a combination with the synergistic bimetallic (Zn/Co) effect. A similar effect was found for ZIF-8 (Table 10). The conversion of ECH improved after the partial substitution of 2-MIM by Atz.

Lee et al. [26] stated that the catalytic behavior of ZIFs can be tuned by synthesis. Thus, in CO_2_ cycloaddition to ECH, the activity of ZIF-8 synthesized in aqueous media was higher than that of ZIF-8 synthesized in methanolic media. This difference was attributed to the dissociation of Zn–N bonds in ZIF-8 synthesized in an aqueous medium, i.e., the formation of a larger number of active sites by dissociated N fragments (pyrrolic and pyridinic) and low-coordinated Zn^2+^ ions (without one [Zn–Im_III_] or two [Zn–Im_II_] linkers) in the framework.

In addition to chemical composition, ZIF structure affects the catalytic properties of ZIFs [38]. Figure 5C shows that an increase in pore size (d_p_) increased the conversion of PO. This trend can be explained by the changes in the accessibility of active sites and diffusion limitations. The larger the pore aperture, the larger the accessibility to active sites and the lower diffusion limitation. Therefore, the higher activity of MAF-6 than that of other ZIFs was attributed to the larger aperture of MAF-6 (7.6 Å) than that of the studied catalysts.

## 4. Conclusions

The main objective of our review is to analyze the recent results on the application of ZIFs in acid–base catalysis in terms of their structure–property–activity relationships. We examine a few reactions that represent the broad range of the potentially promising applications of Zn–ZIFs as heterogeneous catalysts. Literature published within the recent 10–12 years suggests that various factors affect the textural, physicochemical, and catalytic properties of ZIFs. An examination of the literature has led to several tentative conclusions:1.Tuning the affinity for polar molecules (hydrophobic/hydrophilic properties) by varying the chemical composition of the linker may be useful for the application of ZIFs in reactions involving polar reagents (water, alcohols, and aldehydes). The linker can act as an active site for the activation of reagents;2.The textural properties of ZIFs (S_BET_ and V_M_/V_y_) can be changed in two ways: replacing Zn^2+^ with Co^2+^ changes and increases the particle size of ZIFs which then leads to an increase in S_BET_ and V_M_/V_y_. The accessibility and number of active sites decrease in this order. These changes may affect the reaction rate and selectivity;3.The number of active sites is fine-tuned on the basis of the chemical composition and particle size of ZIFs. To date, only information on the basicity of ZIFs, which indicates that the strength of BSs depends on the basicity of linkers and the structure of ZIFs, exists;4.ZIFs possess a unique combination of acidic sites and BSs that allow their use in acid–base catalysis. In most reactions, their catalytic behavior is defined by the presence of the LAS–BS pair. In these cases, the nature of the metal ion and the basicity of the organic linker should be considered because these parameters are crucial for the activation of reactants and the subsequent rearrangement of intermediates;5.The effect of particle size has received little attention in the literature. However, the shape and size of particles should be considered because these factors affect textural properties and the active site number. Moreover, particle size can provoke problems in the course of cyclic tests and/or the isolation of ZIFs from reaction mixtures. We expect that those challenges may be addressed by establishing composites or magnetically separable materials on the basis of ZIFs.

## Data Availability

The published articles were analyzed in this review. Supporting Information of these articles can be finding on sites of Journals where these articles are published.

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
