# Peer review of "Insights into the Structure–Property–Activity Relationship of Zeolitic Imidazolate Frameworks for Acid–Base Catalysis"

_ijms, 2023, doi:10.3390/ijms24054370_

Round 1
Reviewer 1 Report
This Review reports the recent literature report on the ZIFs application in acid-base catalysis wrt the structure-property-activity relationship. Specifically, the article highlights the zeolitic imidazolate frameworks (ZIFs), whose structure comprises imidazolate linkers and metal ions (Co2+, Cu2+, Zn2+, etc.). The theme of the review is novel and considering the recent advancements in the field of ZIFs the article can be considered for publication. However, a few minor modifications are required before it can be accepted in IJMS.
1. The language of the article is very difficult to understand, in spite of good content the language errors and sentence formation masks the clarity of the content.
2. The figure resolution must be improved. Fig. 1 and Fig 2 c,d, inset, and legends are not clear.
3. Considering the number of articles available the literature discussion is not sufficient. More literature reports with comparison and contrast results must be included.
4. Recent literature must be included. All the references date back to 2018 or before.
Reviewer 2 Report
I have reviewed the manuscript, in which author reviewed recent literature about zeolite imidazolate frameworks (ZIFs) application for acid-based catalysis using structure-property-activity relationship. Additionally, author described several reactions which shows heterogeneous properties of Zn-ZIFs. Author also talked about all different factors which can affect various textural, physicochemical and catalytic properties of ZIFs.
The work seems to fall in line with the journal’s scope, providing insights on catalytic properties of the ZIFs. All questions asked in the introduction appear to be sufficiently addressed throughout the text. Therefore, I recommend that this manuscript is ready for the publication as it is.
